# Mouse WIF1 Is Only Modified with *O*-Fucose in Its EGF-like Domain III Despite Two Evolutionarily Conserved Consensus Sites

**DOI:** 10.3390/biom10091250

**Published:** 2020-08-28

**Authors:** Florian Pennarubia, Emilie Pinault, Bilal Al Jaam, Caroline E. Brun, Abderrahman Maftah, Agnès Germot, Sébastien Legardinier

**Affiliations:** 1Glycosylation and cell differentiation, PEIRENE, EA 7500, Faculty of Sciences and Technology, University of Limoges, F-87060 Limoges, France; florian.pennarubia@uga.edu (F.P.); emilie.pinault@unilim.fr (E.P.); bilal.al.jaam@hotmail.com (B.A.J.); caroline.brun@univ-lyon1.fr (C.E.B.); agnes.germot@unilim.fr (A.G.); sebastien.legardinier@unilim.fr (S.L.); 2Complex Carbohydrate Research Center, University of Georgia, Athens, GA 30602, USA; 3Mass Spectrometry Platform, BISCEm, US 042 INSERM-UMS 2015 CNRS, Faculty of Medecine and Pharmacy, University of Limoges, F-87025 Limoges, France; 4NeuroMyoGene Institute, CNRS UMR 5310, INSERM U1217, University of Claude Bernard Lyon 1, 69008 Lyon, France

**Keywords:** click chemistry, EGF-LD, *O*-fucosylation, phylogeny, Pofut1, Wif1

## Abstract

The Wnt Inhibitory Factor 1 (Wif1), known to inhibit Wnt signaling pathways, is composed of a WIF domain and five EGF-like domains (EGF-LDs) involved in protein interactions. Despite the presence of a potential *O*-fucosylation site in its EGF-LDs III and V, the *O*-fucose sites occupancy has never been demonstrated for WIF1. In this study, a phylogenetic analysis on the distribution, conservation and evolution of Wif1 proteins was performed, as well as biochemical approaches focusing on *O*-fucosylation sites occupancy of recombinant mouse WIF1. In the monophyletic group of gnathostomes, we showed that the consensus sequence for *O*-fucose modification by Pofut1 is highly conserved in Wif1 EGF-LD III while it was more divergent in EGF-LD V. Using click chemistry and mass spectrometry, we demonstrated that mouse WIF1 was only modified with a non-extended *O*-fucose on its EGF-LD III. In addition, a decreased amount of mouse WIF1 in the secretome of CHO cells was observed when the *O*-fucosylation site in EGF-LD III was mutated. Based on sequence comparison and automated protein modeling, we suggest that the absence of *O*-fucose on EGF-LD V of WIF1 in mouse and probably in most gnathostomes, could be related to EGF-LD V inability to interact with POFUT1.

## 1. Introduction

Wnt Inhibitory Factor 1 (Wif1), like Cerberus and members of secreted Frizzled-related protein (sFRP), is an extracellular antagonist of both canonical and non-canonical Wnt signaling pathways [1]. Indeed, it can bind to Wnt proteins and prevents them from interacting with the cysteine-rich domain of the Frizzled receptor [2]. The canonical Wnt/β-catenin pathway is essential during vertebrate embryonic development [3] and in homeostasis of almost all adult tissues [4,5]. Therefore, the deregulation of this signaling pathway is frequently associated with human diseases and cancers [6,7].

Wif1 was first identified in human retina and then isolated from mouse, *Xenopus* and zebrafish [8]. It contains an N-terminal signal peptide, a β-sandwich WIF domain [9], five 31-33 amino acid-long EGF-like domains (EGF-LDs), each containing six cysteines connected by three conserved disulfide bonds and a hydrophilic C-terminus [10]. The WIF domain was shown to confer an inhibitory activity to Wif1 [8] but the presence of EGF-LDs I-V was necessary for full activity by strengthening Wif1 binding to Wnt proteins [10]. In addition, these EGF-LDs were shown to bind to negatively charged heparan sulfate proteoglycans (HSPGs) through electrostatic interactions with positively charged residues of EGF-LDs II-IV to regulate Wnt morphogen gradients [10].

WIF1 belongs to the hundred membrane or secreted proteins, 99 found in human and 92 in mouse [11], which are potentially modified with *O*-fucose due to presence of the consensus *O*-fucosylation motif C^2^XXXX(S/T)C^3^ (where C^2^ and C^3^ are the second and third conserved cysteines, respectively) [12] within at least one of their EGF-LDs. Among them, only a few mammalian proteins have been confirmed to be modified with *O*-fucose such as NOTCH receptors [13,14] and its DELTA and JAGGED ligands [15], tissue-plasminogen activator (t-PA) [16] and urokinase-plasminogen activator (u-PA) [17], blood coagulation factors (VII, IX, XII) [18,19,20], AGRIN [21], AMACO [22], CRIPTO-1 [23] and VERSICAN [24]. Recently, we demonstrated that PAMR1, a secreted protein associated to muscle regeneration [25], was modified with *O*-fucose in its unique EGF-LD [11].

The *O*-fucosylation of EGF-LDs-containing proteins is mediated by the endoplasmic-resident protein *O*-fucosyltransferase 1 (POFUT1) [26], which is widely distributed in animals [27]. Its presence is highly correlated with *O*-fucosylable EGF-LDs of the human EGF type (hEGF-LDs), characterized by a C^5^–C^6^ loop with eight or nine residues [28], such as those found in Wif1 [29]. The characterization of hEGF-LDs binding to POFUT1 revealed three main regions involved, namely the C^1^–C^2^, C^2^–C^3^ and C^5^–C^6^ loops of hEGF-LDs as well as their unique residue found between C^4^ and C^5^ [29]. More precisely, the C^2^–C^3^ loop, which includes the *O*-fucosylation motif, was shown to be composed of residues establishing one sulphur-hydrogen and several hydrogen bonds with highly conserved residues of POFUT1, located in a deep groove between the two Rossmann-fold domains [29,30]. The C^1^–C^2^ loop plays a minor role in substrate binding in contrast to the C^5^–C^6^ loop and the residue at position C^4^+1, which modulate substrate-binding affinity through apolar interactions. Thus, the addition of *O*-fucose results in correct positioning of hEGF-LDs in a large solvent exposed pocket of POFUT1, connected to a more buried conserved cavity accommodating the GDP-fucose as a donor substrate [30].

To date, the presence of *O*-fucose was associated with few biological roles. The *O*-linked fucose is widely involved in regulation of interactions between Notch receptor and its membrane-bound Delta and Jagged/Serrate ligands [13,31] but also in AGRIN functions such as aggregation of acetylcholine receptors [21]. Other functions such as a role in protein secretion were clearly attributed to the *O*-fucosylation mediated by POFUT2 for some proteins containing thrombospondin type 1 repeats (TSR) such as ADAMTS13 [32] and the matricellular protein CCN1 [33] but it was not clearly demonstrated for proteins modified with *O*-fucose added by POFUT1.

Furthermore, POFUT1-mediated *O*-fucosylation of Notch receptor involves several levels of regulation since some of its *O*-fucoses can be elongated with *N*-acetylglucosamine (GlcNAc) by enzymes of Fringe family (Lunatic, Manic and Radical). The extension of some *O*-fucoses by Fringe is involved in the positive or negative modulation of Notch interactions with its ligands Delta and Jagged [34,35,36].

Using a phylogenetic approach, we focused on the distribution and evolutionarily conservation of Wif1 as of its potential *O*-fucosylation sites in metazoans. In gnathostomes, we identified two potential conserved *O*-fucosylation sites in EGF-LDs III and V, whereas in protostomes only the first site was sporadically found. In gnathostomes, we showed that WIF1 harbored a highly conserved *O*-fucosylation site on its EGF-LD III while the *O*-fucosylation sequence was more divergent in EGF-LD V. Given the difficulties in isolating natural WIF1, a first approach was performed to determine the ability of recombinant isolated EGF-LDs III and V of mouse WIF1 to be modified with *O*-fucose using click chemistry and mass spectrometry, as previously described [37]. We thus demonstrated for the first time that only isolated EGF-LD III of mouse WIF1 could carry an *O*-fucose. This result was confirmed for full-length WIF1 from CHO cells exhibiting an *O*-fucose in its EGF-LD III, which can be in vitro recognized by recombinant Lunatic Fringe (LFNG). Finally, we showed that the loss of *O*-fucose reduced secreted amount of mouse WIF1 from stably transformed CHO cells.

## 2. Materials and Methods

### 2.1. Phylogenetic Reconstruction and Sequence Conservation Analyses

Wif1 orthologs were retrieved from GenBank (https://www.ncbi.nlm.nih.gov) database using on-site tblastn, blastp and psi-blast facilities, with mouse WIF1 sequence (NP 036045.1) as query. The collected homologous sequences (n > 300) were aligned with MUSCLE [38] implemented in SeaView v.4 [39]. Alignment is available upon request. Only complete sequences covering the maximum taxonomic diversity were selected (n = 47) and 301 homologous sites were retained using Gblocks [40]. Phylogenetic analyses were performed with maximum likelihood (ML) method using PhyML v.3.0 [41], the LG empirical amino acid substitution matrix [42] and gamma-distribution (Γ) of among-site rate variation (4 discrete categories) [43] and estimated proportion of invariant sites and with Bayesian phylogenetic inference using PhyloBayes v4.1c [44], LG + Γ evolution model associated with a category (CAT) mixture model [45], which accounts for across-site heterogeneities in the amino-acid replacement process. Two independent runs were conducted with a total length of 20,000 cycles. They were compared to check the convergence of continuous parameters of the models and assess the convergence in tree space. They satisfactorily converged (maxdiff less than 0.032). The 1000 initial trees were discarded as burn-in and the majority-rule posterior consensus tree was computed from the remaining sub-sampled trees to collect posterior probabilities. For the ML tree, non-parametric bootstrap proportions were calculated after 500 replicates. The determination of the exon-intron organization of mouse WIF1 was obtained using ncbi online utility Splign [46] with Wif1 mRNA (NM 011915.2) and chromosome 10 DNA (NC 000076.6). Multiple alignment portions from the entire dataset were used to obtain Wif1 logos by the TEXshade package [47]. Subfamily logos [48] were then generated for gnathostome (n = 25) and protostome sequences (n = 22), focusing on EGF-LDs III and V. Edition of phylogenetic trees and selection of color portions of the alignment were obtained by MEGA6 [49] and Bioedit v.7.2.5 [50], respectively. Percentages of similarity were calculated online at http://www.bioinformatics.org/sms2/ident_sim.html.

### 2.2. Plasmid Constructs

We cloned mouse WIF1 (NP_036045.1, residues 29–379) cDNA into the pSecTag/FRT/V5-His-TOPO^R^ (pSec vector) (Thermo Fisher Scientific, Waltham, MA, USA), in order to obtain a secreted protein with C-terminal V5 and polyhistidine tags (V5-His). WIF1 counterparts mutated for *O*-fucosylation sites (T255A, T319A, T255/319A) were generated using the GENEART Site-Directed Mutagenesis System (Thermo Fisher Scientific). Mouse POFUT1 (NP_536711.3, residues 31–389) without its endogenous signal peptide and without its C-terminal KDEL-like motif (RDEF) was previously cloned in a modified pSec vector (named pSec-NtermHis6) containing a secretory signal peptide fused to a polyhistidine tag [37]. Then, a V5 tag was added at the N-terminus downstream the His6 tag. The beta-1,3-*N*-acetylglucosaminyltransferase lunatic Fringe (LFNG) (NP_032520.1, residues 86–378) cDNA without its signal peptide and without the N-terminal region known to be cleaved by a furin-like protease was also cloned between *Kpn*I and *Bam*HI unique restriction sites into the pSec-NtermHis. A V5 tag was also added at the KpnI site to produce a secreted protein with N-terminal His and V5 tags. For isolated WIF1 EGF-LDs, cDNAs encoding mouse WIF1 amino acids 243-275 (EGF-LD III) (WT and T255A) and 307-339 (EGF-LD V) (WT and T319A) with or without T/A mutations were cloned between *Bam*HI and *Xho*I into the pET-25b(+) vector (Novagen, Millipore, Burlington, MA, USA), using the same cloning technique by complementary oligonucleotides hybridization as our previous study [37]. Similarly, amino-acids 983-1,019 (EGF-LD 26) of mouse NOTCH1 (N1) (NP_032740.3) and its mutant counterpart on its *O*-fucosylation site T997A were previously cloned into the pET-25b (+) vector [37]. All the sequences of pET- and pSec-derived constructs were verified, used to transform BL21 bacteria and to stably transfect mammalian cells, respectively.

### 2.3. Protein Expression and Purification

Recombinant mouse WIF1 (WIF1-V5-His), POFUT1 (recPOFUT1) and LFNG (recLFNG) were expressed by stable Flp-In^TM^ adherent CHO cells (Thermo Fisher Scientific, Waltham, MA, USA). After production during 72–96 h in serum-free Opti-MEM I medium (Thermo Fisher Scientific, Waltham, MA, USA), proteins were recovered by centrifugation at 1000× *g* for 5 min from cell culture supernatants. Then, proteins were concentrated by several centrifugation steps at 3900× *g* for 45 min at 4 °C in binding buffer (25 mM Tris-HCl, 500 mM NaCl, 5 mM CaCl_2_, 20 mM imidazole, pH 7.5) using Amicon ultra centrifugal filters 10K (Millipore, Burlington, MA, USA) and purified on the Ni-NTA column by imidazole gradient (from 20 to 500 mM imidazole) using AKTA prime system (GE Healthcare, Piscataway, NJ, USA). Isolated EGF-LDs for mouse WIF1 and NOTCH1 were produced in BL21 bacteria after 4 h incubation at 37 °C in LB broth supplemented with 100 μg mL^−1^ ampicillin and 1 mM IPTG. After lysis with 0.5 mg mL^−1^ lysozyme and sonication, soluble proteins were recovered in supernatants after centrifugation at 10,000× *g* for 20 min at 4 °C and diluted in binding buffer before being purified in the same way. All recombinant purified proteins were concentrated with Amicon ultra centrifugal filters 3K (Millipore, Burlington, MA, USA) in the same buffer (25 mM Tris, 5 mM CaCl_2_, pH 7.5) and quantified using a bicinchoninic acid (BCA) protein assay (Sigma-Aldrich Corp. St. Louis, MO, USA) with bovine serum albumin as a standard. Reverse-phase HPLC of Ni-NTA-purified EGF-LDs III and V were then performed on a C18 column with an acetonitrile:H_2_O gradient running from 20% to 50% (*v*/*v*) in the presence of 0.06% (*v*/*v*) trifluoroacetic acid, as previously described [29,37].

### 2.4. Glycosyltransferase Reactions

Before mass spectrometry analyses, 2.5 µg recPOFUT1 were incubated with 5 µg isolated EGF-LD and 0.1 mM GDP-fucose in 20 µl of reaction buffer (25 mM Tris, 5 mM CaCl_2_, 10 mM MnCl_2_, pH 7.5) and incubated overnight at 37 °C as previously described [37]. Additional experiments were carried out with 5 µg isolated EGF-LD III incubated with 2.5 µg of each of the recombinant enzymes (recPOFUT1, recLFNG) and 0.1 mM of each appropriate nucleotide sugar (GDP-fucose, UDP-GlcNAc) in 20 µL of reaction buffer as above.

Before click chemistry experiments, in vitro glycosyltransferase reactions were carried out with 1 µg recPOFUT1 mixed with 2 nmoles GDP-azido-fucose (R&D Systems Inc., Minneapolis, MN, USA) and 2 µg isolated EGF-LD or 1 µg recombinant WIF1 protein variant in 25 µL reaction buffer and incubated overnight at 37 °C. For *O*-fucose extension with GlcNAc, 1 µg recLFNG was mixed with 2 nmoles UDP-azido-GlcNAc (R&D Systems Inc., Minneapolis, MN, USA) and 1 µg recombinant WIF1 protein variant for overnight incubation at 37 °C before being subjected to click chemistry.

### 2.5. Click Chemistry Reactions

As previously described [37], copper-assisted azide–alkyne cycloaddition (CuAAC) was performed using 1.25 mM CuCl_2_, 2.5 mM ascorbic acid and 0.125 mM alkynyl biotin (R&D Systems Inc., Minneapolis, MN, USA), directly added to the glycosyltransfease reaction. The mixture was incubated in the dark for 1 h at room temperature and stopped by heating for 5 min in Laemmli buffer [51].

### 2.6. SDS-PAGE and Blotting Techniques

Crude recombinant proteins from culture media were recovered by centrifugation at 1000× *g* for 5 min. Protein from cell pellets were incubated on ice for 30–60 min with RIPA extraction buffer (50 mM Tris-HCl, 150 mM NaCl, 0.5% sodium deoxycholate, 1% NP-40, 0.1% SDS, pH 8) containing a protease inhibitor cocktail (Roche Applied Science, Mannheim, Germany) and soluble proteins were recovered in the supernatant after centrifugation at 14,000× *g* for 15 min. Proteins were then quantified using a bicinchoninic acid (BCA) protein assay (Sigma-Aldrich Corp., St. Louis, MO, USA). Crude or purified proteins were separated on polyacrylamide gels (SDS-PAGE), which were either stained with silver nitrate or blotted to nitrocellulose membranes (amido black staining can be used to reveal transferred total proteins). After blocking with 5% non-fat milk in Tris-buffered saline/Tween-20 (TBST) (50 mM Tris–HCl, 150 mM NaCl, 0.1% Tween-20, pH 7.4) for 1 h at room temperature, membranes were incubated overnight at 4 °C with V5 Tag monoclonal antibody (Thermo Fisher Scientific, R961-25, Waltham, MA, USA ) or anti-GAPDH (R&D Systems Inc, AF5718, Minneapolis, MN, USA) diluted to 1:2000 in 2.5% non-fat milk-TBST. For anti-GAPDH, membranes were incubated after three washes in TBST for 1 h at room temperature with 1:2000 dilution of appropriate secondary HRP conjugate antibodies (Dako, Glostrup, Denmark) in 2.5% non-fat milk-TBST. For experiments with click chemistry, samples were separated as described above and transferred to membranes. Then, membranes were blocked with 10% non-fat milk-TBST for 10 min and incubated with streptavidin-HRP (Thermo Fisher Scientific, 434323, Waltham, MA, USA) in TBST at 25 ng/mL for 30 min. The membranes were washed three-times before and after streptavidin-HRP incubation with TBST (15 min per wash). In both cases, membranes were revealed using enhanced chemiluminescence peroxidase substrate. Signals were visualized and quantified using Amersham Imager 600 (Cytiva, Marlborough, MA, USA).

### 2.7. Automatic Modeling and Superimposition with X-ray Structures

Automatic homology models were generated for mouse WIF1 EGF-LD III and EGF-LD V on the Swiss-model server (https://swissmodel.expasy.org), using the X-ray structure of human WIF1 (PDB 2YGQ) [10] as a reference template, since both EGF-LD III and V shared more than 40% identity with the human protein. More precisely, mouse WIF1 EGF-LDs III and V exhibited 96.88% and 40.63% identity with human WIF1 EGF-LD III and EGF-LD I (human EGF-LD V was not crystallized [10]), respectively. Other templates proposed for EGF-LD V with the best scores of identity (54.84%) and Global Model Quality Estimation (GMQE) (0.84) were dismissed because of the presence of an additional residue in the C^2^–C^3^ loop. They were considered as relevant structural models in view of their GMQE, 0.98 for EGF-LD III and 0.74 for EGF-LD V. Using MatchMaker of UCSF CHIMERA [52], mouse N1-EGF-LD 26, co-crystallized with murine POFUT1 (PDB 5KY4) [29], was superimposed with obtained models for EGF-LD III and V. Finally, it was replaced with either EGF-LD III or EGF-LD V at the same location in murine POFUT1 to identify potential steric clashes and charge repulsions.

### 2.8. Protein Digestions, Mass Spectrometry Analyses and Data Processing

Glycosyltransferase reactions in 25 mM ammonium bicarbonate were reduced in 5 mM dithiothreitol, alkyled in 10 mM iodoacetamide and digested overnight at 37 °C using 0.1 µg of trypsin (Promega Corp., Madison, WI, USA) alone or combined with 0.1 µg thermolysin in 0.5 mM CaCl_2_ (Promega Corp., Madison, WI, USA). The peptide samples were then purified on 1CC 30 mg HLB cartridge (Waters Corporation, Milford, MA, USA) with the following steps: conditioning with 1 mL methanol, equilibration with 0.5% formic acid in water, loading of sample diluted in 0.5% formic acid in water, 2 washes with 0.5% formic acid in water and elution with 1 mL methanol. After evaporation under nitrogen flow, the digests were resolubilized in 50 µL of loading mobile phase (water/acetonitrile/trifluoroacetic acid (98/2/0.05%)) and finally filtered on 0.22 µm spin column (Agilent Technologies, Santa Clara, CA, USA).

Resulting peptides were analyzed by microLC-MS/MS using a nanoLC 425 in micro-flow mode (Eksigent, Dublin, CA, USA) system coupled with a TripleTOF 5600+ (SCIEX, Framingham, MA, USA). Five µl of each sample was trapped on a C18 PepMap 100 cartridge (300 µm ld × 5 mm, 5 µm; Thermo Fisher Scientific, Waltham, MA, USA) and desalting was carried out at 10 μL/min with Loading mobile phase for 5 min. Chromatographic separation was performed on a ChromXP C18 column (150 × 0.3 mm i.d., 120 Å, 3 µm; SCIEX) at a flow rate of 3 µL/min. The mobile phase was a gradient of water/acetonitrile/formic acid 100/0/0.1% (A) and 5/95/0.1% (B) programed as follows: initial, 5% B, increased to 25% over 20 min, then increased to 95% B over 2 min, maintained at 95% for 4 min and finally, decreased to 5% B for re-equilibration. The TripleTOF 5600+ was operated in information-dependent acquisition (IDA) mode with Analyst TF 1.7 software (SCIEX). Mass spectrometry (MS) and tandem mass spectrometry (MS/MS) data were continuously recorded with up to 20 precursors selected for fragmentation from each MS survey scan. Precursor selection was based upon ion intensity and whether or not the precursor had been previously selected for fragmentation (dynamic exclusion). Collision energies were automatically adjusted to the charge state and *m*/*z* value of the precursor ions.

The recombinant protein sequence database was searched with ProteinPilot 5.0 (SCIEX) and the Multiple reaction monitoring (MRM) transition list was established using Skyline 3.5.0 (MacCoss Lab, University of Washington, Seattle, WA, USA) for the WT and mutated non-modified peptides. *O*-fucosylation (coded by [dHx] amino acid modification) was added in silico at the expected position with PeakView software 2.2 (SCIEX) and *m*/*z* of precursor and fragments were calculated. Data were acquired in high-Resolution MRM (MRM^HR^) mode: product ion scans were collected for the *m*/*z* corresponding to WT and mutated peptides, with or without *O*-fucose modification, during 30 min using the same parameters as previously described in the Information-Dependent Acquisition (IDA) method (Appendix A). Data were processed with MultiQuant Software 3.0.1 (SCIEX), considering the six most abundant fragments for each peptide with a resolution of 10,000. The same fragments were used for non-modified peptides and those modified with *O*-fucose (Appendix A). Areas were collected for the same most major fragment of non-modified peptide and those carrying *O*-fucose: a percentage of *O*-fucosylation was calculated.

As in our recent study [11], precursor *m*/*z* corresponding to all possible combinations of glycosylated peptides were also calculated and searched in previous analyses leading to MS1 identification when the error between theorical and observed *m*/*z* was less than 10 ppm. *m*/*z* of peptides with an elongated *O*-linked fucose were also used to create a MRM^HR^ method targeting peptides with all combinations of elongated structures. MRM^HR^ data were processed with the same fragments as described above (MS2 identification).

### 2.9. Statistical Analysis

All experiments were performed in biological triplicates and results were reported as the means ± SEMs. Statistical comparisons were performed using two-tailed *t* tests implemented in Prism, version 5.03 (GraphPad Software, Inc., San Diego, CA, USA). A *p* value of 0.05 or less was considered statistically significant.

## 3. Results

### 3.1. Wif1 Appeared in Bilaterian Ancestor

More than 300 Wif1 orthologs were retrieved from diverse databases using tblastn, blastp and psi-blast programs of the BLAST^®^ algorithm. No sponge or diploblastic species were identified with the characteristic WIF domain and the 5 EGF-LDs. We selected 47 complete sequences, 27 from deuterostomes (exclusively represented by sequences of gnathostomes because only partial sequences were retrieved from databases for earlier emerging taxa) and 20 from protostomes, in order to encompass maximal bilaterian taxonomic diversity. Their sizes varied from 373 amino acids (aa) for *Ornithorhynchus anatinus* to 381 for *Astyanax mexicanus* in deuterostomes and from 353 aa for *Onthophagus taurus* to 456 for *Drosophila melanogaster* (due to a longer N-terminal part) in protostomes. The Maximum likelihood (ML) tree (Figure 1) clearly separated protostomes from deuterostomes and supported our current knowledge of animal evolutionary relationships [53].

Monophylies of amniotes, sarcopterygians, lophotrochozoans and arthropods were supported by bootstrap proportions of 88%, 86%, 70% and 100%, respectively. Bayesian inference of the phylogenetic tree produced the same topology, except for the teleost paraphyly and their earlier emergence relative to chondrichthyes. This was surprising, considering that the site-heterogeneous CAT mixture model used is able to overcome LBA (Long Branch Attraction) artefacts [54]. LBA phenomenon causes systematic errors in phylogeny as it clusters sequences based on their shared dissimilarity (due to mutational saturation of sites) relative to closely related groups of organisms and consequently does not reveal their true evolutionary relationships. Percentages of similarity of the selected deuterostome and protostome sequences ranged from 71.1% (*Astanyax* vs. *Ornithorhynchus*) to 96.6% (*Camelus* vs. *Orycteropus*) and from 33.8% (*Drosophila* vs. *Mizuhopecten*) to 88.4% (*Megachile* vs. *Microplitis*), respectively. The distinct degrees of dissimilarity resulted in significantly longer branches for protostomes compared to deuterostomes, underlying different selective pressures and functional divergences as demonstrated for *Drosophila* Wif1 ortholog, Shifted and human WIF1, targeting Hedgehog or Wingless/Wnt morphogens, respectively [55].

Among partial sequences, Wif1 was found in the early diverging deuterostome taxon of ambulacrarians, with the echinoderms *Strongylocentrotus purpuratus* (XP_003724586.1, XP_011670169.1, XP_783155.2) and *Lytechinus variegatus* (JI441084) and the hemichordate *Saccoglossus kowalevskii* (NP_001161492.1). Surprisingly, *Strongylocentrotus purpuratus* was the only species for which several different Wif1 sequences were recovered from databases. No Wif1 ortholog was found in cephalochordates, urochordates, cyclostomes, platyhelminths or nematodes.

### 3.2. Predicted O-Fucosylation Sites Are Evolutionarily Conserved in Deuterostomes

The consensus sequence C^2^XXXX(S/T)C^3^ for *O*-fucosylation was searched in the complete and partial Wif1 sequences (Figure 2).

The consensus sequence of *O*-fucosylation was found in EGF-LDs III and V for nearly all gnathostomes (Appendix A) but not in EGF-LD V for the nine-banded armadillo *Dasypus novemcinctus* and the squamate *Gekko japonicus*. It is also absent in EGF-LD III for the American pika *Ochonta princeps* and the platypus *Ornithorhynchus anatinus,* but interestingly, this latter exhibited another potential *O*-fucosylation site (C^2^RNGGSC^3^) in its EGF-LD I (Figure 2). Surprisingly, the snakes *Python bivittatus* and *Thamnophis sirtalis* were the only deuterostome representatives for whom Wif1 was devoid of *O*-fucosylation sites. In protostomes, the consensus site was mainly found in EGF-LD II and sometimes in EGF-LD III and IV but never in EGF-LD V (Appendix A). Furthermore, in the partial sequences of early emerging deuterostomes, the ambulacrarians (echinoderms and hemichordates), a consensus site was also found in EGF-LD II (Figure 2). The most parsimonious explanation concerning the evolution of *O*-fucosylation sites among the 5 EGF-LDs of Wif1 is that during bilaterian evolution, EGF-LD II was the first to contain an *O*-fucosylation motif. In deuterostomes, after ancestral emergence of ambulacrarians, it was replaced by those of EGF-LDs III and V in gnathostomes.

In protostomes, most had conserved the ancestral situation with the *O*-fucosylation site in EGF-LD II, with in some cases an additional site in EGF-LD III or IV for spiralians (mollusks and annelids). The site in EGF-LD II could have disappeared in favor of sites in EGF-LDs III and IV for brachiopods such as *Lingula anatina* or was never replaced in some hexapods as springtails (*Folsomia*) and endopterygotes (*Drosophila, Tribolium*).

*Wif1* gene, located on chromosome 10 D2-D3 in *Mus musculus*, is composed of 10 exons encoding a 379 aa protein whose EGF-LDs are encoded by a 96-bp exon each (Figure 3).

The organization with 5 EGF-LDs and each one encoded by one exon was probably ancestral in bilaterians since it was also found in other gnathostomes (*Homo sapiens*, *Ornithorhynchus anatinus*, *Taeniopygia guttata*, *Xenopus tropicalis*, *Danio rerio*), in hemichordates (with the partial sequence of *Saccoglossus kowalewskii*) and in annelids (*Capitella teleta*). Genomic sequences of protostomes, such as *Acyrthosiphon pisum* and *Drosophila melanogaster,* showed a single exon for EGF-LDs I and II and for EGF-LDs IV and V. Interestingly, the only splicing sites conserved in bilaterians were those bordering the exon encoding EGF-LD III. The potential *O*-fucosylation site in EGF-LD III of *Mus musculus* WIF1 was C^2^FNGGTC^3^ and corresponded to the consensus site widely found in gnathostomes. The second site, present in EGF-LD V, was C^2^GAHGTC^3^ but it was less conserved in gnathostomes. When EGF-LD III sequences were compared between gnathostomes and protostomes (Figure 3), 12 homologous sites were significantly different: T^255^ in deuterostomes vs. K in protostomes but also some amino acids present between C^1^–C^2^, C^3^–C^4^ and C^4^–C^5^. Ten sites were different concerning EGF-LD V, mostly present between C^1^–C^2^, C^3^–C^4^ and on the last 7 positions of the EGF-LD.

### 3.3. Only WIF1 EGF-LD III Can Be In Vitro Modified by O-Fucose

For analysis of the *O*-fucosylation status of the natural WIF1 protein, preliminary tedious and time-consuming steps are required to specifically enrich or purify this very low abundant secreted glycoprotein from organisms such as mouse before performing mass spectrometry analysis. To bypass these difficulties, recombinant proteins for full-length protein or its isolated EGF-LDs were produced, purified and characterized. Thus, we first determined the propensity of recombinant mouse WIF1 EGF-LDs III and V purified from *E.coli* BL21 strain to be specifically in vitro modified with *O*-fucose by recombinant POFUT1 (recPOFUT1). As previously reported [11,37,56], *O*-fucosylation assays were followed either by click chemistry (CuAAC) and blotting technique or by trypsin digestion and mass spectrometry to specifically reveal and/or quantify EGF-LDs *O*-fucosylation (Figure 4).

Recombinant WT EGF-LDs III and V were first assayed for their ability to receive in vitro *O*-fucose, using NOTCH1 EGF-LD 26 (N1-EGF-LD 26) as a previously confirmed positive control [37]. T/A mutated counterparts for each EGF-LD were used as negative controls. After independent incubations of each EGF-LD with recPOFUT1 and GDP-azido-fucose, click chemistry (CuAAC) was used to bind biotin alkyne to azido fucose after being attached to an EGF-LD by *O*-linkage (Figure 4A, scheme). After SDS-PAGE (sodium dodecyl sulfate-polyacrylamide gel electrophoresis) and a blotting technique, streptavidin-HRP was then used to reveal biotinylation of transferred *O*-fucoses. For mouse WIF1 EGF-LD III, a positive signal around 12 kDa was only obtained for WT counterpart, consistent with its successful modification with azido *O*-fucose and at the predicted position, T^255^ (Figure 4A, upper panel). For mouse WIF1 EGF-LD V counterparts, WT and T319A, no signal was detected. As expected, N1-EGF-LD 26 was efficiently modified with azido-fucose by recPOFUT1. Indeed, a strong signal around 12 kDa corresponding to monomers was revealed and a second slight signal was detected near 25 kDa that could correspond to the presence of EGF-LD dimers (also slightly visible for WT EGF-LD III). All these signals were considered as specific because T/A mutated counterparts were undetected, despite the same protein quantities were loaded on gel, as shown by the Coomassie blue-stained polyacrylamide gel (Figure 4A, lower panel).

Slight differences in apparent MW were observed between EGF-LDs, which could be attributed to differences in composition of charged amino acids and not to defects in signal peptide cleavage. Indeed, full-scan liquid chromatography with tandem mass spectrometry (LC-MS/MS) analyses were performed and measurements of deconvoluted MW of undigested EGF-LDs III and V were correlated with EGF-LDs lacking signal peptides (Appendix A). All folding isomers for EGF-LDs III and V preparations from IPTG-induced *E.coli* BL21 strain were separated by RP-HPLC and analyzed separately (Appendix A). After in vitro *O*-fucosylation assay for all eluted proteins and click chemistry, we also showed that EGF-LD III could be modified with *O*-fucose unlike EGF-LD V.

To confirm these results and quantify in vitro *O*-fucosylation, multiple reaction monitoring-mass spectrometry (MRM-MS) was carried out as previously described [37]. After independent incubation of WT EGF-LDs III and V of mouse WIF1 with recPOFUT1 and GDP-fucose as a donor substrate, trypsin digestion followed by MRM-MS were performed. Trypsin-digested peptides, containing the potential *O*-fucosylation consensus sites T^255^ and T^319^, were generated for EGF-LDs III and V respectively. For EGF-LD III, peaks corresponding to 18 residue-long peptides with or without *O*-fucose modification were both detected (Figure 4B, upper panels). The rate of *O*-fucosylation for EGF-LD III was 67.24% ± 1.80% (n = 3). For WT EGF-LD V, only the non-modified 28 residue-long peptide was found (Figure 4B, lower panels). The same experiment was done with the same amounts of WIF1 EGF-LDs III and V mutated on threonines (T255A and T319A) and no peptide with *O*-fucose was detected (Appendix A). Altogether, these results showed that only WIF1 EGF-LD III was prone to be modified in vitro by POFUT1 with a fucose specifically transferred to T^255^.

### 3.4. Full-Length Recombinant WIF1 Carried O-Fucose Only on Its EGF-LD III

The full-length recombinant mouse WIF1 (WIF1-V5-His), produced in secretome of stable CHO cell line, was purified and first analyzed by MRM-MS after trypsin digestion as for isolated EGF-LDs but the peptides of interest, modified or not, were not detected (data not shown). Failure in detection of the generated peptide for EGF-LD III (A(N^245^)CSTTCFNGG(T^255^)CFYPGK) could be attributed to the presence of the *N*-glycan carried by N^245^. For EGF-LD V, the peptide generated by trypsin digestion, different by 8 residues at the N-terminus (GYQGDLCSKPVCEPGCGAHG(T^319^)CHEPNK) compared to that obtained for isolated EGF-LD V (Figure 4B), was also not detected by mass spectrometry. To overcome these problems, MRM-MS analysis was performed after co-digestion by trypsin and thermolysin. These new digestions generated smaller peptides of interest, namely with the sequence FNGGT^255^C^3^ for EGF-LD III and the sequence VC^1^EPGC^2^GAHG(T^319^)C^3^HEPNK for EGF-LD V after a missed cleavage by thermolysin (Figure 5A). This new strategy is thus based on identification of peptides with the same sequences whether we performed co-digestion of full-length WIF1 or its isolated EGF-LDs. To validate this new strategy, trypsin/thermolysin co-digestion and MRM-MS analysis were first performed for recombinant EGF-LD III after incubation with recPOFUT1 and GDP-fucose and the same results were obtained as for trypsin single digestion (Appendix A).

Consistent with results obtained for isolated EGF-LDs and with similar rate of *O*-fucosylation, WIF1-V5-His was found to be endogenously modified with *O*-fucose on T^255^ of EGF-LD III, as revealed by MRM-analysis after protein co-digestion (Figure 5B, upper panels). However, a small percentage of molecules was not modified, suggesting that endogenous expression of POFUT1 in CHO cells was not sufficient to modify 100% of WIF1-V5-His molecules with *O*-fucose. Similar results were obtained in our previous study focusing on PAMR1, another POFUT1 target protein, produced in CHO cell lines [11]. Interestingly, incubation of WIF1-V5-His with recPOFUT1 and GDP-azido-fucose led to successful fucose transfer to T^255^ of these molecules, which were not endogenously modified with *O*-fucose by CHO cells as attested by click chemistry experiments associated with blotting technique (Appendix A). So the incomplete *O*-fucosylation of recombinant WIF1-V5-His on its EGF-LD III was due to insufficient level expression of endogenous POFUT1 in these cells compared to WIF1-V5-His overexpression.

For EGF-LD V, only the non-modified peptide of interest was detected (Figure 5B, lower panels), consistent with our result related to inability of isolated EGF-LD V (Figure 4) and WIF1-V5-His T255A (Appendix A) to be in vitro modified by recPOFUT1.

Despite this successful strategy, trypsin/thermolysin co-digestion can only be used for preparations of purified WIF1 and not for a complex protein extract since the peptide FNGGTC is not only found in WIF1. Indeed, it is also found in other POFUT1 target proteins such as NOTCH receptors where it can also be modified or not with *O*-fucose.

### 3.5. Full-Length Recombinant WIF1 Carried a Non-Extended O-Fucose on Its EGF-LD III

The *O*-fucose detected on the EGF-LD III of WIF1-V5-His, produced by CHO cells, is susceptible to be extended with other monosaccharides to form *O*-linked di-, tri- and/or tetra-saccharides. MRM-MS analysis was performed to detect all possible combinations but no combination for *O*-fucose elongation was detected (Appendix A). Lunatic Fringe (LFNG) is a golgian glycosyltransferase known to specifically extend *O*-linked fucose attached to an EGF-LD by transferring a *β*-d-GlcNAc residue from UDP-D-GlcNAc [57]. To know if the *O*-fucose carried by T^255^ could be specifically extended with GlcNAc by LFNG, EGF-LD III was first co-incubated with recPOFUT1, recLFNG and donor nucleotide sugars (GDP-fucose, UDP-GlcNAc) (Figure 6).

After trypsin/thermolysin co-digestion and MRM-MS, the digested peptide FNGGT^255^C^3^ of EGF-LD III was found to be mainly modified with *O*-fucose and to a lesser extent with the *O*-linked disaccharide GlcNAc-Fuc (Figure 6A). The click chemistry approach was then used to determine if WIF1-V5-His endogenously modified with non-extended *O*-fucose by CHO cells could be a substrate for LFNG. After in vitro GlcNAcylation using recLFNG and UDP-azido-GlcNAc followed by click chemistry (CuAAC) associated with blotting techniques using streptavidin-HRP, specific signals were detected at expected MW only for WT and T319A WIF1-V5-His (Figure 6B scheme and upper panel). This result showed successful LFNG-mediated addition of azido-GlcNAc to *O*-fucose carried by T^255^ of WIF1-V5-His produced by CHO cells (Figure 6B, upper panel). A second signal appearing just above 35 kDa may correspond to LFNG which remains bound to untransferred azido-GlcNAc, as it was previously reported for *O*-GlcNAc transferase (OGT) in case of GlcNAc transfer, using a similar click chemistry approach and azido nucleotide sugar [58]. On the contrary, T255A and T255/319A mutated proteins were not revealed after in vitro incubation with recLFNG, consistent with the fact that T^319^ was not endogenously modified with *O*-fucose by CHO cells. For these proteins, LFNG labelling was never seen despite loading of same protein quantities and qualities as for WT and T319A (Figure 6B, lower panel). It could be due to the inability of T255A and double T/A mutant to form a stable ternary complex with recLFNG and UDP-azido-GlcNAc.

### 3.6. The O-Fucose Carried by T^255^ Was Required for Optimal Secretion of Recombinant WIF1

POFUT2-mediated *O*-fucosylation was shown to be involved in secretion of several TSR-containing proteins such as ADAMTS13 [32] but it was not clearly demonstrated for proteins modified with *O*-fucose by POFUT1 such as WIF1. Therefore, we wondered if loss of *O*-fucosylation sites could impact WIF1-V5-His secretion. Using anti-V5-HRP antibody, relative quantifications were performed by Western Blot on WT and mutant WIF1 proteins expressed by CHO cells (Figure 7).

Pellets and supernatants of culture media were recovered and analyzed to discriminate between secretion modifications and differences in protein expression. When analyzing same volumes of culture medium (conditioned media) loaded on polyacrylamide gels for each recombinant protein, a significant decrease in quantity of more than 50% was seen for WIF1-V5-His mutated on its *O*-fucosylation site T^255^ (T255A) or the double mutant T255/319A compared to WT proteins. However, secretion of T319A WIF1 was not statistically different from that of WT (Figure 7A), consistent with occupancy of the *O*-fucosylation site T^255^ only. All these results on conditioned media were inversely correlated with those obtained for cell pellets (Figure 7B), for which the proteins devoid of *O*-fucose modification, namely T255A WIF1 and the double mutant, were more retained in the intracellular compartment. We can thus hypothesize that presence of *O*-fucose on WIF1-V5-His could influence the efficiency of its intracellular trafficking and secretion.

## 4. Discussion

In this paper, we showed that recombinant mouse WIF1-V5-His carried *O*-fucose only on T^255^ of its EGF-LD III when it was produced in CHO cells, similarly to the unique EGF-LD of recombinant mouse PAMR1 that we also produced in these cells [11]. The high conservation of the sequence C^2^XNGGTC^3^ in gnathostomes strongly suggests that Wif1 EGF-LD III could be also occupied by *O*-fucose in other species (Appendix A). However, the C^2^–C^3^ loop of EGF-LD V, which, according to our results, does not carry *O*-fucose, was more divergent in gnathostomes with the following sequence C^2^GX(H/Y)G(S/T)C^3^. H or Y at C^2^+3 could be responsible for major steric clashes and perturbations of vicinal hydrogen bonds as previously suggested [29]. In addition, the arginine in position C^5^+1 of EGF-LD V was probably involved in another steric clash and/or charge repulsion in *Mus musculus*, which could affect binding to POFUT1. The situation was similar in other gnathostomes, for which the arginine residue was replaced by glutamine or another basic residue (lysine or histidine) (Appendix A). All these observations could reflect inability of POFUT1 to correctly position EGF-LD V in the binding cavity for fucose transfer, whatever the considered species of gnathostomes.

Strikingly, we showed in this study that the presence of *O*-fucose added on EGF-LD III by POFUT1 was required for optimal secretion of recombinant mouse WIF1. Among gnathostomes, some species (platypus and ambulacrarians such as *Saccoglossus kowalevskii*) unusually possess an *O*-fucosylation motif in EGF-LD I or EGF-LD II respectively, instead of EGF-LD III (Figure 2). We can assume that the probable negative effect of loss of the *O*-fucosylation site in EGF-LD III on Wif1 secretion might be offset by the gain of a new *O*-fucosylation site (C^2^XNGGTC^3^) in EGF-LD I or EGF-LD II found for these species. In the same way, we can speculate that a total lack of *O*-fucose could lead to diminished Wif1 secretion in snakes (*Python bivittatus*, *Thamnophis sirtalis*), in which Wif1 is devoid of *O*-fucosylation site and in the pika *Ochotona princeps,* which only possess one *O*-fucosylation consensus sequence located in EGF-LD V. We can also think that Wif1 secretion may be not affected in these species due to compensatory effects reducing the requirement for *O*-fucosylation on EGF-LD III such as interactions with some protein partners promoting secretion and/or a different global glycan composition. A large diversity of protostomes only has one potential *O*-fucosylation motif located in EGF-LD II (Figure 2). Among protostomes, *Limulus polyphemus* and three arthropods (*Centruroides sculpturatus*, *Daphnia pulex* and *Cryptotermes secundus*) (Figure 1) exhibited in their EGF-LD II the most similar *O*-fucosylation sequence (C^2^MNGG(S/T)C^3^) to that found in EGF-LD III of gnathostomes (Appendix A). This observation is a good indicator of a possible modification of this EGF-LD by an O-fucose. It allows us to highlight species of protostomes where the detection of *O*-fucose by click chemistry on Wif1 orthologs deserves to be characterized.

To understand the ability of WIF1 EGF-LD III to carry an *O*-fucose contrary to mouse WIF1 EGF-LD V, automated homology models were generated using X-ray structure of human WIF1 [10] as a reference template. Using MatchMaker of CHIMERA [52], the generated model obtained for EGF-LD V was first superimposed with N1 EGF-LD 26 co-crystallized with mouse POFUT1 [29] and with a relevant model for EGF-LD III, based on the known 3D structure for human WIF1 with its first three EGF-LDs [10]. The overall shape of models for EGF-LDs III and V was comparable to that of N1-EGF-LD 26, exhibiting correctly superimposed C^2^–C^3^ and C^5^–C^6^ loops, known to be involved in interactions with POFUT1 (Figure 8A, left panel).

These EGF-LDs should have thus the same correct positioning in the binding pocket of POFUT1 (Figure 8A, right panel), if side-chains of residues that belong to the most important region for interaction with POFUT1, namely the C^2^–C^3^ loop, are similar. It was only the case of N1-EGF-LD 26 and EGF-LD III, exhibiting the sequence C^2^FNGGTC^3^, allowing us to replace N1-EGF-LD 26 by WIF1 EGF-LD III in the complex with mouse POFUT1 (Figure 8B, left panel). Considering POFUT1 interactions with mouse NOTCH1 EGF-LDs 12 and 26 [29], EGF-LD III in mouse and most gnathostomes having the *O*-fucosylation sequence C^2^XNGGTC^3^ should *a minima* establish three important links with mouse POFUT1 to allow *O*-fucosylation. Indeed, EGF-LD III is likely to establish a sulphur-hydrogen bond between its C^3^ residue and POFUT1 M^46^ as well as two hydrogen bonds, between its G^254^ and T^255^ residues and POFUT1 G^47^ and N^51^, respectively. The N^252^ of EGF-LD III could thus also form hydrogen bond with N^151^ of POFUT1.

In addition, other identical or similar amino acids, outside the C^2^–C^3^ loop and known to be involved in interactions with POFUT1, were found in the same positions such as a serine residue of the C^1^–C^2^ loop interacting with POFUT1 R^138^ and the aliphatic residue in position C^4^+1 interacting with a group of apolar residues on POFUT1. All these interactions could be responsible for correct positioning of EGF-LD III in the deep groove of POFUT1, leading to a subsequent transfer of fucose. Based on superimposition of X-ray crystal structure of mouse POFUT1 in complex with mouse EGF-LD 26 [29] with automatic structural models, Wif1 EGF-LD II of the protostome *Daphnia pulex*, having the *O*-fucosylable sequence C^2^MNGGSC^3^, might be modified with *O*-fucose by its own Pofut1 (Appendix A), suggesting conservation of Wif1 biological activities during bilaterian evolution.

On the contrary, the replacement of N1-EGF-LD 26 with automated model for EGF-LD V revealed several inconsistencies regarding a potential interaction with mouse POFUT1 (Figure 8B, right panel), such as major steric clash between the voluminous residue H^317^ of C^2^–C^3^ loop (C^2^GAHGTC^3^) and Y^78^ of POFUT1. It could lead to a lack of interaction with mouse POFUT1, it would be likewise for other gnathostome Wif1 where a tyrosine is found at the homologous position in EGF-LD V (Appendix A). Indeed, it was recently shown that an aspartate at this position (C^2^QNDATC^3^) in NOTCH1 EGF-LD 12 was responsible for a steric clash leading to a weak interaction with POFUT1 [29]. Furthermore, we recently showed that the in vitro ability of POFUT1 to transfer fucose was significantly lower for NOTCH1 EGF-LD 12 compared to EGF-LD 26 [37]. Thus, it is tempting to speculate that an amino acid with a bulky side chain at this position could systematically decrease and even prevent interaction with POFUT1.

Outside the C^2^–C^3^ loop of EGF-LD V, another steric clash and/or potential charge repulsions between the R^329^ at position C^5^+1 and POFUT1 R^48^ might take place. In addition, residues known to reduce binding affinity [29] such as a proline in the C^1^–C^2^ loop (strictly conserved in gnathostomes) and glutamine at the C^4^+1 position (largely distributed in gnathostomes) were found in EGF-LD V. All these residues could explain the inability of WIF1 EGF-LD V to be modified with *O*-fucose by POFUT1 in mouse and probably in gnathostomes despite presence of the consensus *O*-fucosylation motif C^2^XXXX(S/T)C^3^. Indeed, POFUT1 is not able to distinguish between hEGF-LDs having an *O*-fucosylable S or T residue and those without one [29]. Thus, the nature of residues in an EGF-LD must be considered, namely residues within the C^2^–C^3^ loop and to a lesser extent, those at the C^4^+1 position and in the C^5^–C^6^ loop.

In this study, our results gave strong evidence that mouse WIF1 EGF-LD III was modified with *O*-fucose on its T^255^ unlike EGF-LD V, which could be unable to interact with POFUT1 despite its evolutionarily conserved *O*-fucosylation consensus sequence. Interestingly, we showed that the *O*-fucose carried by T^255^ of WIF1-V5-His could be in vitro extended with GlcNAc by recLFNG but no *O*-fucose extension was detected by MRM-MS for WIF1-V5-His, probably due to a too low expression of endogenous Lunatic Fringe in CHO cells. It was also the case for mouse PAMR1, which exhibited a non-extended *O*-fucose on its unique EGF-LD when produced in CHO cells [11]. We can still wonder if mouse WIF1 can be a natural target for other GlcNAc transferases of the Fringe family (Lunatic, Manic and Radical) [36] and if the *O*-linked GlcNAc-fucose disaccharide might be extended to form a sialylated *O*-fucosylglycan.

## 5. Conclusions

Finally, we showed for the first time a role of EGF-LDs *O*-fucosylation in protein secretion in mammals. It would be interesting to study other potential effects for *O*-fucosylation of WIF1, particularly the ability of WIF1 bearing *O*-fucose to interact with Wnt proteins and/or with HSPGs [10]. Indeed, *O*-fucosylglycans are known to contribute to protein–protein interactions such as those between Notch receptors and DSL ligands [35,59,60]. It would also be relevant to examine the potential role of extendable *O*-fucose by Fringe in the modulation of biological activities of WIF1, as demonstrated for Notch [57,61].

## Figures and Tables

**Figure 1 biomolecules-10-01250-f001:**
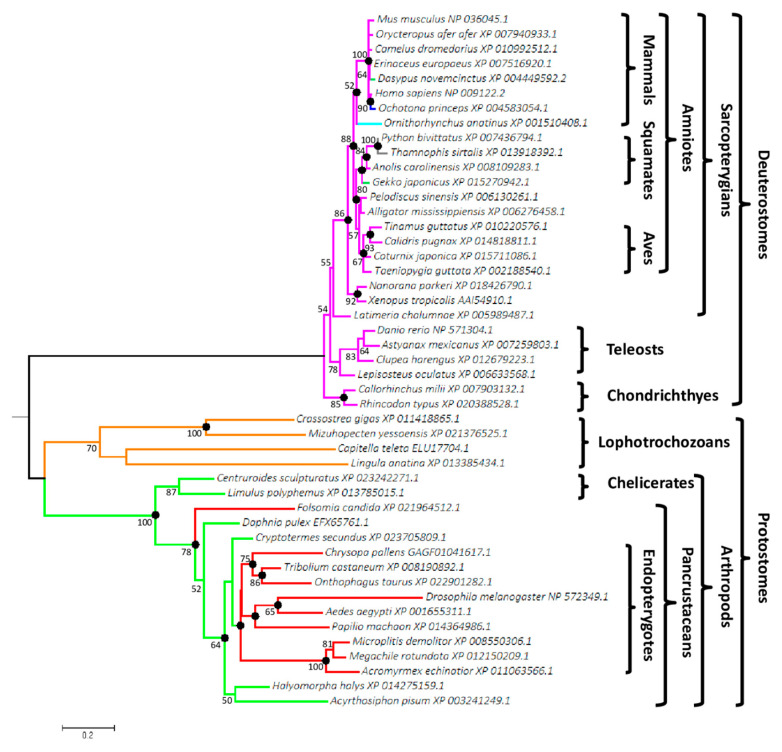
Phylogenetic tree of bilaterians based on Wif1 comparison. The tree was reconstructed from 47 species and 301 aligned positions using the PhyML method with LG + Γ (4 rates) evolution model. The best maximum likelihood (ML) tree had a log-likelihood (LnL) value of −10,048.71 and the estimated value of α shape parameter of the discrete Γ distribution was 1.35. The tree was drawn to scale and mid-point rooted. The scale bar represents the number of substitutions per site. Non-parametric bootstrap percentages from ML analysis (500 replicates) appeared for nodes when ≥ 50%. Filled circles indicated nodes with estimated posterior probabilities >0.95 in Bayesian inference. Genbank accession numbers are indicated after the species name. Branch colors represent monophyletic groups or species, which differed in the number and position of Wif1 potential *O*-fucosylation sites.

**Figure 2 biomolecules-10-01250-f002:**
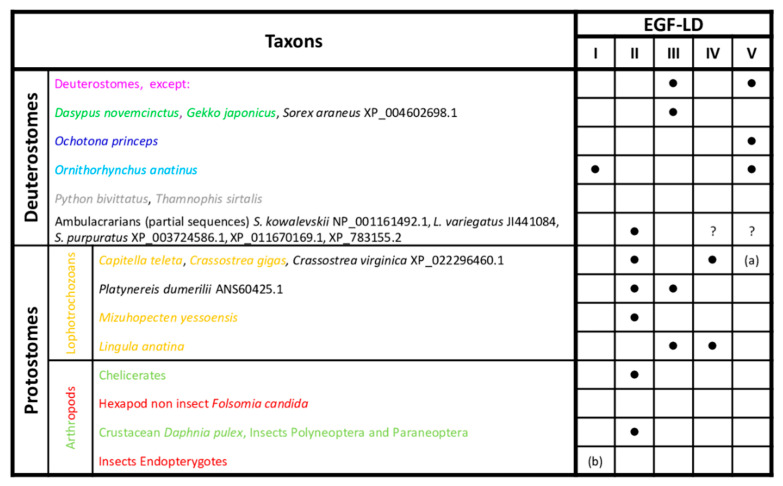
Conservation of potential Wif1 *O*-fucosylation sites in bilaterians. Presence of the consensus sequence, C^2^XXXX(S/T)C^3^, for *O*-fucosylation in deuterostomes and protostomes is indicated by a black dot. The different font colors of species correspond to those of branches shown in Figure 1. Group and species in black font were not included in the phylogenetic reconstruction. (**a**) EGF-LD V absent in *Crassostrea spp.* (**b**) EGF-LD I absent in Hymenoptera. ? EGF-LDs IV and V absent due to partial sequences.

**Figure 3 biomolecules-10-01250-f003:**
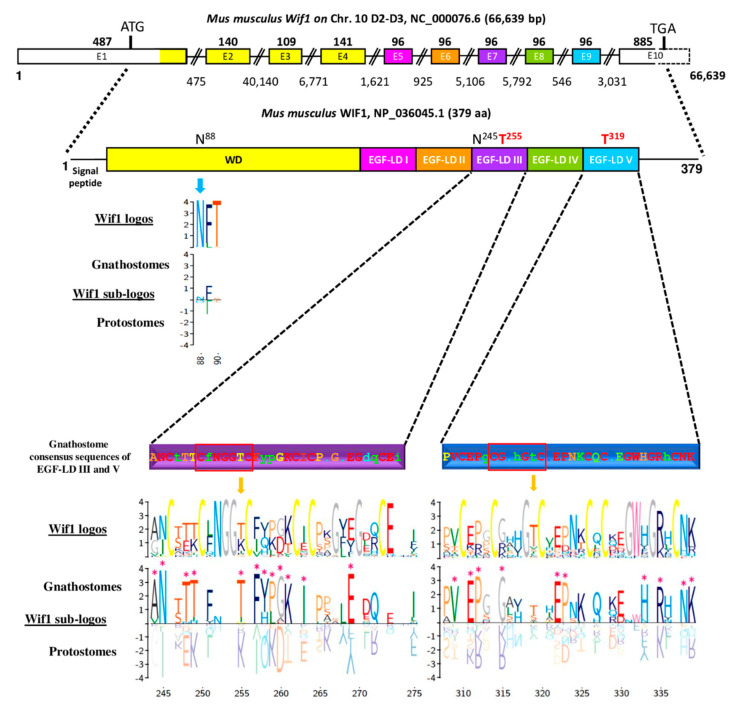
Mouse Wnt Inhibitory Factor 1 (Wif1) (gene structure and modular protein) and sequence comparison between gnathostomes and protostomes focusing on EGF-like domains (EGF-LDs) III and V. Gene structure of *Mus musculus Wif1* on chromosome 10 with above numbers corresponding to exon sizes (E1 to E10) and below, to intron sizes. Only exon 10, partially bordered by a dotted line, was not drawn to scale. The protein sequence contained a signal peptide, a Wif domain (WD), five EGF-LDs (I to V) and a C-terminal hydrophilic domain. Colors of the different domains are indicated on their corresponding coding exons. Consensus sequence of gnathostomes for predictive *O*-fucosylation (T^255^, T^319^) sites, homologous to those present in mouse WIF1, are distinguished according to a HotCold color variation and in uppercase for identity ≥80%, in lowercase, between 50% and 80% and with a dot, when <50%. Logos for Wif1 orthologs and subfamily logos comparing gnathostomes to protostomes were created. Numbering is according to the mouse WIF1 sequence. The height of the letters represents the amino acid relative frequency at each position. Sub-family logos display relevant deviations (*) of a sub-family compared to the other. The location of the two predicted *N*-glycosylation consensus sites (N88 and N245) in mouse WIF1 is indicated for protein sequence.

**Figure 4 biomolecules-10-01250-f004:**
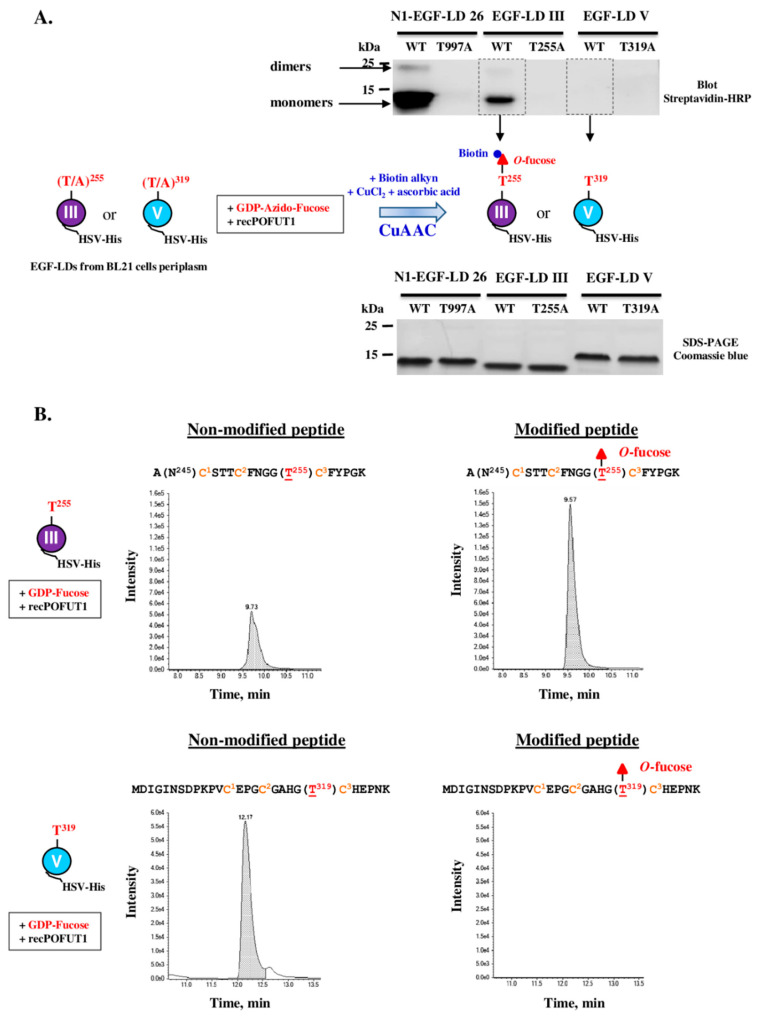
In vitro *O*-fucosylation assay with WIF1 EGF-LDs III and V followed either by click chemistry and blotting technique or by trypsin digestion and mass spectrometry. (**A**) Isolated WT EGF-LDs III and V or T/A mutated on T^255^ and T^319^ respectively were first incubated with recPOFUT1. Then, azido-labeled GDP-fucose and click chemistry (CuAAC) was performed using alkynyl biotin to covalently attach biotin to fucose (red filled triangle) if transferred to an EGF-LD by recPOFUT1. After separation by SDS-PAGE and protein transfer, protein biotinylation was detected using streptavidin-HRP. Mouse NOTCH1 EGF-LD 26 (N1-EGF-LD 26), known to be modified with *O*-fucose in mouse NOTCH1, was used as a positive control. Positive signals resulted from successful *in* vitro *O*-linked azido-fucose transfer to EGF-LDs (upper panel), for which quantity and purity were checked by Coomassie blue-stained polyacrylamide gels (lower panel). (**B**) EGF-LD III and EGF-LD V, produced and purified from *E. coli* BL21 strain, were first independently incubated with recPOFUT1 and GDP-fucose to induce in vitro *O*-fucosylation. After reduction, alkylation and trypsin digestion, resulting peptides were analyzed by micro-LC multiple reaction monitoring-mass spectrometry (MRM-MS). Non-modified peptides were detected for EGF-LDs III and V (left panels) but the peptide modified with *O*-fucose was only revealed for EGF-LD III (right panels). The amino acid sequence of peptide generated by trypsin for each WIF1 EGF-LD is indicated with its *O*-fucosylation site in red and in brackets. Conserved cysteines (in orange) are numbered.

**Figure 5 biomolecules-10-01250-f005:**
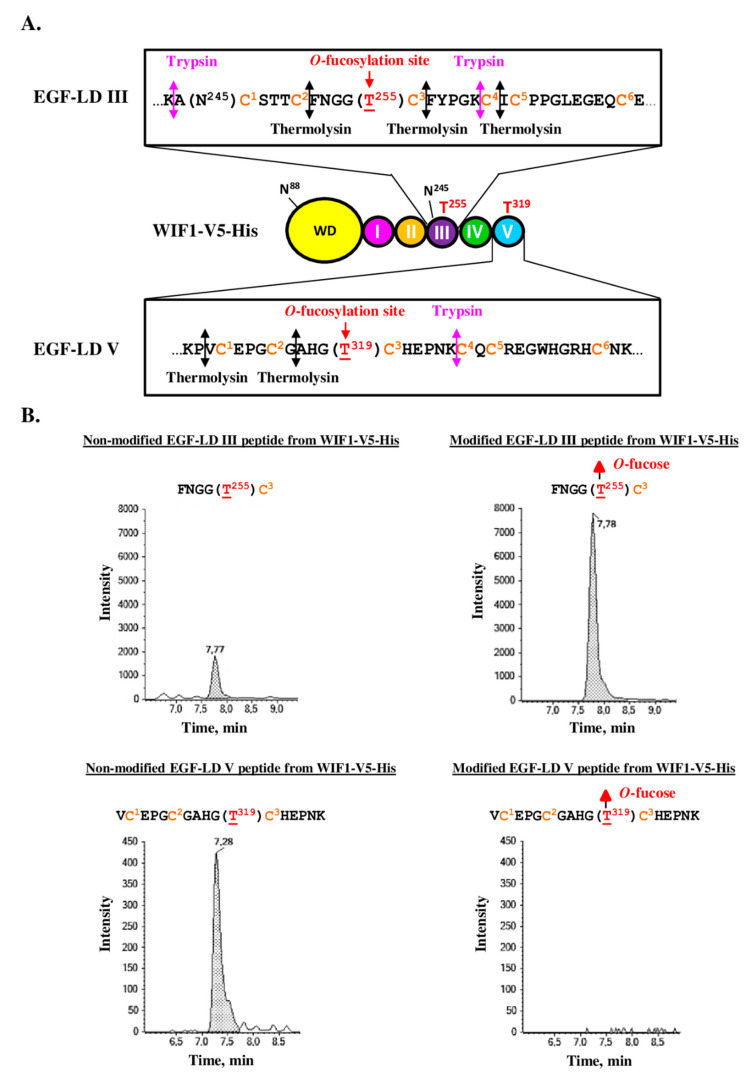
MRM-MS of co-digested WIF1-V5-His, produced by stable CHO cell lines. (**A**) The recombinant mouse WIF1 with its C-terminal V5 and His tags (WIF1-V5-His) is drawn with its different domains. Zooms on the amino acid sequence of its EGF-LDs III and V are boxed and show the protease cleavage sites by trypsin and thermolysin. (**B**) Full-length recombinant mouse WIF1-V5-His, secreted in culture medium of stable CHO cell lines, was purified, reduced, alkylated and finally co-digested by trypsin and thermolysin. Resulting peptides were analyzed by micro-LC MRM-MS. Non-modified peptides (left panels) were detected for both EGF-LDs but only EGF-LD III peptide was detected with *O*-fucose modification (right panels).

**Figure 6 biomolecules-10-01250-f006:**
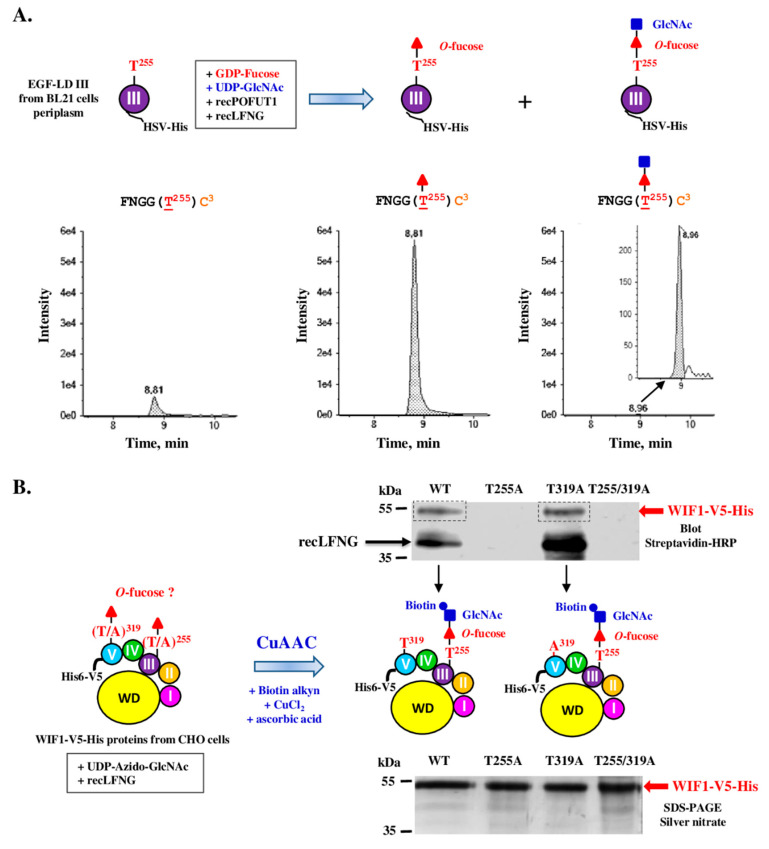
In vitro *O*-fucose elongation assay with EGF-LD III or WIF1-V5-His proteins followed either by trypsin/thermolysin co-digestion and mass spectrometry (**A**) or by click chemistry and blotting technique (**B**). (**A**) EGF-LD III, produced and purified from *E. coli* BL21 strain, was co-incubated with recPOFUT1, recLFNG and appropriate donor nucleotide sugars (GDP-fucose, UDP-GlcNAc) to induce in vitro *O*-fucosylation and subsequent extension with GlcNAc. After reduction, alkylation and trypsin/thermolysin digestion, the peptide of interest FNGGT^255^C^3^ of EGF-LD III, which was analyzed by micro-LC MRM-MS, was mainly modified with *O*-fucose (*O*-Fuc) and also modified with the *O*-linked disaccharide GlcNAc-Fuc but at a lesser extent. (**B**) WT and mutated (T255A, T319A and T255/319A) recombinant mouse WIF1-V5-His, purified from stable Flp-In^TM^ CHO cells, were subjected to incubation with recLFNG and UDP-Azido-GlcNAc. Then, click chemistry (CuAAC) was performed using alkynyl biotin to covalently attach biotin to GlcNAc if transferred to *O*-fucose carried by WIF1-V5-His. After separation by SDS-PAGE and protein transfer, protein biotinylation was detected using streptavidin-HRP. Positive signals corresponded to successful in vitro LFNG-mediated azido-GlcNAc transfer to recombinant WIF1 proteins (upper panel), for which quantity and purity were checked by silver nitrate-stained polyacrylamide gels (lower panel). RecLFNG, which remains bound to untransferred azido-GlcNAc, appeared labelled around 40 kDa after incubation with WT or T319A WIF1.

**Figure 7 biomolecules-10-01250-f007:**
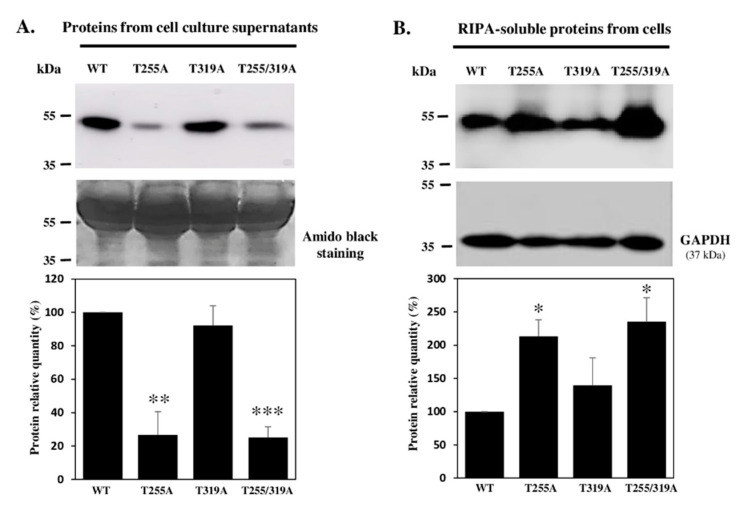
Immunodetection of recombinant mouse WT and T/A mutated WIF1-V5-His proteins, secreted in culture medium or retained by cells. Soluble proteins from cell culture supernatants (**A**) and extracted with RIPA buffer from stable Flp-In^TM^ CHO cells (**B**) were analyzed by Western blot using anti-V5-HRP antibody to reveal WT WIF1-V5-His and mutated forms on *O*-fucosylation sites (T255A, T319A and T255/319A). Histograms in (**A**) represent the secretion for mutated proteins compared to WT WIF1. Supernatants recovered from confluent cultures were analyzed, after elimination of floating cells and adjustment up to same total volumes (conditioned media), by protein transfer on nitrocellulose membrane and immunoblot (upper panel) or amido black staining (lower panel). Histograms in (**B**) correspond to the protein within cells (calculated from V5 to GAPDH signal ratios) compared to WT WIF1. For histograms, WT was set to 100% for all three replicates. Mean ± SEM are shown (*t*-test two-tailed, *: *p* < 0.05, **: *p* < 0.01, ***: *p* < 0.001).

**Figure 8 biomolecules-10-01250-f008:**
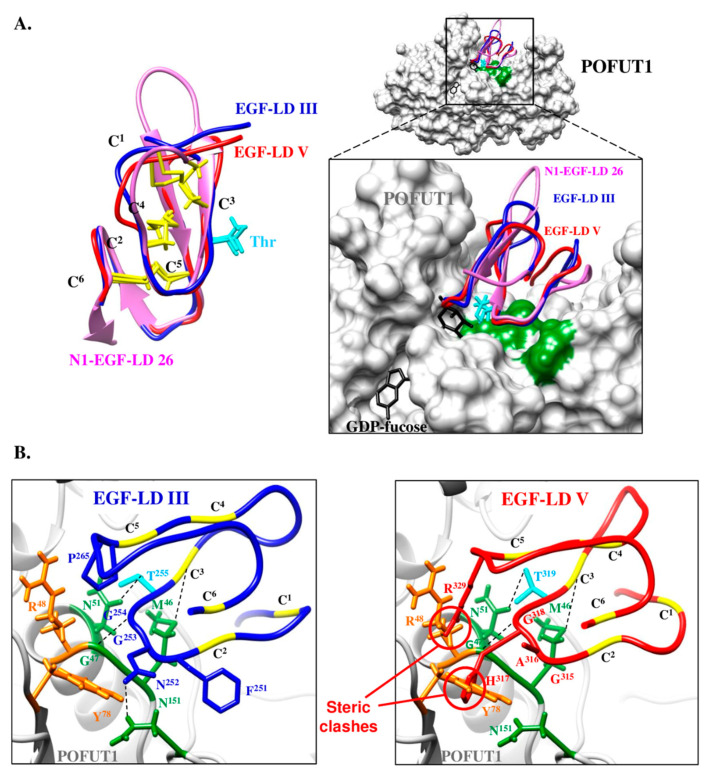
Automated homology models for mouse WIF1 EGF-LD III and EGF-LD V and their interactions with mouse POFUT1. (**A**) Homology models were generated by using Swiss-model Server for mouse WIF1 EGF-LD III (blue) and EGF-LD V (red) using X-ray structure of human WIF1 (PDB 2YGQ) as a reference template. Using Matchmaker of CHIMERA, these models were superimposed with mouse N1 EGF-LD 26 (pink) alone (left) or in complex with mouse POFUT1 (PDB 5KY4) and GDP-fucose (black) (right). The comparison of the three EGF-LDs structures (left), partly based on the conservation of the three disulfide bonds (C^1^–C^3^, C^2^–C^4^, C^5^–C^6^) (yellow), shows similar C^2^–C^3^ and C^5^–C^6^ subdomains and a more divergent C^1^–C^2^ loop. The threonine side chain of the *O*-fucosylation consensus motif is shown in cyan. The inset shows the zoomed interaction region, with POFUT1 residues in green interacting with residues of the EGF-LD C^2^–C^3^ subdomains. (**B**) Substitution of N1 EGF-LD 26 either by EGF-LD III (left) or EGF-LD V (right) in mouse POFUT1 (PDB 5KY4) is shown. Hydrogen and sulphur-hydrogen bonds (schematized by dashes) between key residues (green) of mouse POFUT1 and those of the C^2^–C^3^ subdomain might be unaffected in EGF-LD III since this EGF-LD can be modified with *O*-fucose. However, steric clash and/or charge repulsion between EGF-LD V and residues in orange (R^48^, Y^78^) could explain the inability of POFUT1 to add *O*-fucose to EGF-LD V.

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
