# Peer review of "Mouse WIF1 Is Only Modified with O-Fucose in Its EGF-like Domain III Despite Two Evolutionarily Conserved Consensus Sites"

_biomolecules, 2020, doi:10.3390/biom10091250_

Round 1
Reviewer 1 Report
The authors present an interesting phylogenetic comparison of WIF1 EGF-LD O-fucosylation sites and identify conserved POFUT1 target sites in EGF-LD III and V of gnathostomes. They used a combination of approaches to demonstrate that the mouse EGF-LD III POFUT1 consensus site was modified and extended by FRINGE, and that this O-fucosylation was needed for efficient trafficking of WIF1. In contrast, they found EGF-LD V to be unmodified. Using modeling, they propose that steric hindrance prevents POFUT1 modification of EGF-LD V.
Areas that should be addressed:
507-519 & 512-513 Figure 6B – If lower band is LFNG self-labeling, why was this band not observed in the T255A mutant and T255/319A lanes as LFNG was added to all four samples? Consider alternate interpretation/explanation.
Figure 7 A. Lacks transfection/secretion control or protein loading control.
Figure 7 B. Gel image for anti-V5-HRP is low resolution and cropped too close to bands. Gapdh also cropped to close.
Supplementary Figure 9: What is meant by recPOFUT1 self-labelling (POFUT1 does not have EGF motif – how can it self label?). Consider alternate interpretation for the lower band.
Minor clarifications:
Figure 3: Heat map colors are difficult to see on the purple and light blue backgrounds.
537-540 “All these results on conditioned media were inversely correlated with those obtained for cell pellets (Figure 7B), for which WT WIF1-V5-His was the less retained protein in the intracellular compartment. We can hypothesize that presence of O-fucose on WIF1-V5-His could influence its intracellular trafficking and secretion.” Consider rephrasing to describe the increased retention of the mutants rather than the less retained WT.
539-540 “We can hypothesize that presence of O-fucose on 539 WIF1-V5-His could influence its intracellular trafficking and secretion.” Consider changing could influence its…… to could influence the efficiency of its…..
562-564 “In the same way, we can speculate that a total lack of O-fucose 562 could lead to diminished Wif1 secretion in snakes (Python bivittatus, Thamnophis sirtalis), in which 563 Wif1 is devoid of O-fucosylation site, and in the pika Ochotona princeps, which only possess one 564 O-fucosylation consensus sequence located in EGF-LD V.” Why would it be an advantage to decrease secretion efficiency, consider alternate explanation, for example: could there instead be other compensatory changes that reduce the requirement for O-fucosylation on EGF-LD III?
Author Response
Please find below our responses point by point (in red) to the reviewer’s comments.

Reviewer 2 Report
In the manuscript: “Mouse WIF1 is only modified with O-fucose in its 3 EGF-like domain III despite two evolutionarily 4 conserved consensus sites”, Pennarubia et al. use a combination of phylogenetics and mass spectrometry to elucidate O-fucosylation patterns on EGF-LD repeats of the protein Wnt Inhibitory Factor 1 (Wif1) and link O-fucosylation of EGF-LD III to secretion of Wif1. Overall, the paper is well written and the conclusions are well supported by evidence. Only a few minor comments:
- In figure 2, Arthropods should not be in black font, as the caption states group and species in black font were not included in the phylogenetic analyses and Arthropods were included in that analysis.
- Line 297. “It resulted in significantly longer branches for protostomes…” the pronoun “it” is used without a clear reference, thus it is not entirely clear what “it” refers to. The authors here should make it clear what resulted in significantly longer branches (i.e. the phylogenetic tree)
- Line 316. The authors start the section with a pronoun lacking proper reference: “It was found in EGF-LDs III and V”. Please make it clear what “it” refers to.
- Line 331: please change “would” in the statement “The site in EGF-LD II would have…” to could, as the word would there seems too strong a claim.
- Line 351. “this organization was probably ancestral” it is not entirely clear what “this organization” refers to. Does this refer to the entire organization of the gene (ie the intron/exon structure)?, the protein structure? Do all gnathostomes for instance have 10 exons with the same spacing? The authors should make it clear exactly which aspects of gene/protein organization they propose were likely ancestral.
- Line 447-448: the phrase: “However, a small part of molecules was not modified,” could be rewritten as: “However, a small percentage of molecules were not modified,” for better grammar.
- Lines 566-570: the sentence: “Among protostomes, Limulus polyphemus and three arthropods (Centruroides sculpturatus, Daphnia pulex and Cryptotermes secundus) (Figure 1) exhibited in their EGF-LD II the most similar O-fucosylation sequence (C2MNGG(S/T)C3) to that found in EGF-LD III of gnathostomes (Supplementary Figure 4B), thus conferring to them a propensity to be modified with O-fucose.” This sentence is somewhat speculative, as the authors did not test this sequence. The conclusion of a “propensity to be modified with O-fucose” seems a bit strong.
Author Response

(The authors gave the same response as above.)

Round 2
Reviewer 1 Report
The authors addressed my concerns.